# Automatic Identification of Pangolin Behavior Using Deep Learning Based on Temporal Relative Attention Mechanism

**DOI:** 10.3390/ani14071032

**Published:** 2024-03-28

**Authors:** Kai Wang, Pengfei Hou, Xuelin Xu, Yun Gao, Ming Chen, Binghua Lai, Fuyu An, Zhenyu Ren, Yongzheng Li, Guifeng Jia, Yan Hua

**Affiliations:** 1Guangdong Provincial Key Laboratory of Silviculture, Protection and Utilization, Guangdong Academy of Forestry, Guangzhou 510520, China; wkwildlife@sinogaf.cn (K.W.); luckypro7@163.com (P.H.); xuelin_x@163.com (X.X.); younguycm@gmail.com (M.C.); lbhdahuoji@163.com (B.L.); afy_any@163.com (F.A.); 13339492857@163.com (Z.R.); itachi_li@163.com (Y.L.); 2College of Engineering, Huazhong Agricultural University, Wuhan 430070, China; angelclouder@mail.hzau.edu.cn

**Keywords:** pangolins, deep learning, behavior recognition, temporal relative, attention mechanism

## Abstract

**Simple Summary:**

Researching and developing an automated and intelligent method to monitor the pangolin breeding process can effectively help human observation, analysis, breeding and daily behavior studies of pangolins, and has significant implications for the protection and breeding research of pangolin populations. In this paper, a pangolin breeding attention and temporal relative network (PBATn) was used to monitor and identify the breeding and daily behaviors of pangolins. This study demonstrates that the deep learning system can accurately observe pangolin breeding behavior, rendering it useful for analyzing the behavior of these animals.

**Abstract:**

With declining populations in the wild, captive rescue and breeding have become one of the most important ways to protect pangolins from extinction. At present, the success rate of artificial breeding is low, due to the insufficient understanding of the breeding behavior characteristics of pangolins. The automatic recognition method based on machine vision not only monitors for 24 h but also reduces the stress response of pangolins. This paper aimed to establish a temporal relation and attention mechanism network (Pangolin breeding attention and transfer network, PBATn) to monitor and recognize pangolin behaviors, including breeding and daily behavior. There were 11,476 videos including breeding behavior and daily behavior that were divided into training, validation, and test sets. For the training set and validation set, the PBATn network model had an accuracy of 98.95% and 96.11%, and a loss function value of 0.1531 and 0.1852. The model is suitable for a 2.40 m × 2.20 m (length × width) pangolin cage area, with a nest box measuring 40 cm × 30 cm × 30 cm (length × width × height) positioned either on the left or right side inside the cage. A spherical night-vision monitoring camera was installed on the cage wall at a height of 2.50 m above the ground. For the test set, the mean Average Precision (mAP), average accuracy, average recall, average specificity, and average F1 score were found to be higher than SlowFast, X3D, TANet, TSN, etc., with values of 97.50%, 99.17%, 97.55%, 99.53%, and 97.48%, respectively. The recognition accuracies of PBATn were 94.00% and 98.50% for the chasing and mounting breeding behaviors, respectively. The results showed that PBATn outperformed the baseline methods in all aspects. This study shows that the deep learning system can accurately observe pangolin breeding behavior and it will be useful for analyzing the behavior of these animals.

## 1. Introduction

Pangolins belong to the class Mammalia, order Pholidota, family Manidae, and genus Manis [1]. There are 8 pangolin species worldwide, including the Chinese pangolin (*Manis pentadactyla*), the Indian pangolin (*M. crassicaudata*), and the Malayan pangolin (*M. javanica*) in China [2]. Due to its unique habits, habitat fragmentation, and illegal human trade, the Chinese pangolin is endangered in the wild. With declining populations in the wild, captive rescue and breeding have become one of the most important ways to protect pangolins from extinction [3,4]. Due to the lack of knowledge on the adaptive mechanisms of the specialized feeding, nutritional requirements of different physiological stages and reproductive physiology, artificial rescue and breeding of pangolins are still a worldwide challenge [5]. Studying pangolin population breeding requires monitoring pangolins and understanding the behavioral characteristics of their breeding process. Pangolins are nocturnal animals, and their activities and behaviors are difficult to record through continuous human observation. Researching and developing an automated and intelligent method to monitor the pangolin breeding process can effectively help human observation, analysis, breeding and daily behavior studies of pangolins and has significant implications for the protection and breeding research of pangolin populations.

Early cameras were employed to capture animal behavior for human understanding. For instance, Sun used cameras to document pangolin behavior [6], while Ina and Irene manually observed cows’ and pigs’ crawling and jumping behavior [7,8]. However, manual observation is time-consuming, labor-intensive, inefficient, and error-prone. Since 2000, algorithms have automated animal behavior recognition in videos using video image processing. The Euclidean distance recognized abnormal, heat stress, and drinking pig behaviors with accuracy rates of 80.00%, 90.00%, and 90.70%, respectively [9,10,11]. The elliptical fitting model differentiated pig activities, drinking, mounting, and lying behaviors with accuracy rates of 89.80%, 92.00%, and 93.00–95.00%, respectively [12,13,14]. Optical flow features and SVM (support vector machines) were used for sow nursing recognition, achieving accuracy rates of 94.50% [15]. Optical flow features also recognized cow mounting behavior with 98.30% accuracy [16]. SVM, KELM (kernel-based extreme learning machine), and other kernel function learning machines differentiated pig behaviors (attacking, standing, lying, mounting, and nursing) with accuracy rates of 90.20%, 91.86%, and 94.40% [17,18,19]. This work primarily used manually extracted behavior features, making traditional machine learning techniques inefficient for large datasets, susceptible to environmental noise and light changes, and challenging to apply to new scenarios.

Since 2010, deep learning technology has revolutionized video processing, and the study of deep learning algorithms in animal behavior recognition has gained popularity. In the research of animal behavior recognition algorithms, scholars employ two types of datasets: image data sets and video data sets. Scholars increasingly favor video data sets for training networks. In 2020, Fuentes et al. used a dual-stream I3D network and YOLO v3 to merge temporal and spatial information, achieving an mAP (mean Average Precision) value of 78.80% for YOLO v3. After integrating the dual-stream I3D network, the mAP value increased to 85.60% [20]. In 2020, Li et al. applied the SlowFast network to construct a spatiotemporal convolutional network (PMB-SCN) for the multi-behavior recognition of pigs, attaining an mAP of 96.35% [21]. In 2021, Wang et al. employed CNN and LSTM to recognize sow standing and lying behaviors, achieving an mAP of 90.60% [22]. In 2021, Yang et al. utilized Faster R-CNN and spatiotemporal features to recognize pig mounting behaviors, achieving an mAP of 95.15% [23]. These research works leverage 3D convolutional networks, optical flow convolutional networks, and recurrent neural networks to process video data sets, significantly enhancing the accuracy of recognizing pig and cow behaviors.

In animal behavior recognition, behavior characteristics are closely linked to video positions and neighboring frames’ information. In 2014, the attention mechanism was introduced into neural networks. It adjusts attention for space, channel, and time when processing data, improving feature capture. In 2022, Wang et al. used YOLOv5 and attention mechanisms (SENet and C3GC-3 blocks) for cow-mounting recognition (94.30% accuracy) [24]. Gan et al. recognized weaned piglet behaviors using GCNs and attention (mAP 96.69%) [25]. Gao et al. used OAAR to recognize sika deer behaviors (mAP 97.45%) [26]. Gong et al. added SE-Net modules to GoogLeNet for sika deer recognition (98.92% accuracy) [27]. In 2023, Wang et al. added ECA channel attention to the 3D module for cattle behavior recognition (mAP 98.17%) [28]. Attention mechanisms enhance object detection, recurrent, 3D convolutional, and graph neural networks, adjusting attention for improved recognition accuracy.

Therefore, there is an urgent need for a deep learning-based behavioral recognition model to deeply analyze the expression patterns of the breeding behavior of captive pangolins and develop an automatic behavioral monitoring platform for pangolins. This study shows that the deep learning system can accurately observe pangolin breeding behavior and it will be useful for analyzing the behavior of these animals.

## 2. Materials and Methods

### 2.1. Data Collection

The video collection site for pangolins was the Guangdong Province Guangzhou Wildlife Rescue Center. The pangolin cage area was 2.40 m × 2.20 m (length × width), and a nest box measuring 40 cm × 30 cm × 30 cm (length × width × height) was placed in the left or right position inside the cage. During the pangolin sexual maturity period, healthy and fertile male and female pangolins were selected for mating. The temperature in the cage was controlled at approximately 26.50 °C. The staff fed the pangolins at 16:30 every day, stopped feeding once every 7 days, cleaned the cage daily, and checked the health of the pangolins. Efforts were made to minimize external interference during the breeding period.

As pangolins are usually active at night, a spherical night-vision monitoring camera (T12HV3-IA/POE, Hikvision, Hangzhou, China) was used to capture images. The camera was fixed on the cage wall 2.50 m above the ground, with a downward angle of 45.00° and a horizontal field of view of 114.80°. The collected image size was 1920 pixels × 1080 pixels, and the frame rate was 30 frames/s. Video was collected 24 h a day throughout the experiment and stored in mp4 format on a mobile hard drive. A total of 5 batches of pangolin breeding behavior video data were collected in this experiment, and each batch of video captured the breeding behavior of a pair of pangolins, as shown in Table 1.

### 2.2. Data Set Annotation Definition

The behavior of pangolins is roughly divided into breeding behavior, non-breeding behavior, and external interference. Breeding behavior is further divided into two types: mounting and chasing. Non-breeding behavior is further divided into three types: feeding, activity, and resting, where feeding includes feeding and drinking, while activity includes crawling, standing, and climbing. If a staff member enters the pangolin’s activity area for work, it is defined as external interference.

During the breeding season of pangolins, they usually rest in the nest box around 4 a.m. and become active around 5 p.m. Breeding behavior usually occurs during the active period outside the nest box. The male pangolin chases the female pangolin, then mounts and mates with her. The chasing lasts 1–2 min, and the mounting lasts 10–30 min. During the mounting, mating behavior may sometimes occur and last 1–2 min. The male pangolin wraps his tail around the female’s body as evidence of mating. After the mating action, the pangolin stops mounting. There is also a rare phenomenon of female pangolins chasing male pangolins within the cage. When the pangolins are active outside the nest box, they smell the food and climb to the feeding trough to start feeding and drinking, which lasts 5–10 min. After feeding, the pangolins sometimes crawl or stand in the cage, and sometimes climb the nest box. The activity behavior is continuous in the cage. As staff members enter the pangolin’s activity area for daily care work, their behavior is defined as external interference in the data set.

### 2.3. Data Set Making

During the data set annotation process, the method of behavior priority (breeding precedence, BP) labeling is used. Breeding behavior, such as mounting and chasing, is first labeled, followed by non-breeding behavior and external interference. Since breeding behavior is produced by two mating pangolins, one pangolin is usually in the process of feeding or activity during non-breeding behavior. When both breeding behavior and non-breeding behavior occur in a video, breeding behavior is prioritized. Using the BP method to label helps the network model improve sensitivity to breeding behavior. In the study, the method of manually intercepting the single behavior video of the pangolin in the video according to the behavior classification is first used to extract the video, with each video ranging from 8 s to 32 s. Due to the slow movement of pangolins and the small movement between adjacent frames in the video, the method of intercepting images by taking 1 frame out of every 32 frames is used for each video, and the resolution of the image frames is 414 × 240 as the input image set for the model. Then, each image set of pictures is placed in a separate folder labeled for behavior, and the labeling results are shown in Figure 1.

### 2.4. Data Set Structure

In this experiment, a total of 11,476 video segments, with a total duration of 6060.35 min and 363,622 frames, were intercepted. The numbers of video segments and intercepted image frames for each behavior are shown in Table 2. The number of intercepted image frames for each video segment ranges from 8 to 32, and each image set intercepted from each video segment is considered a dataset. To visually compare the recognition results of the model, a test set was randomly selected from the overall image set, as shown in Table 3.

### 2.5. Behavior Recognition Model Structure

To address the uniqueness of pangolin behavior, this study established a temporal relative attention mechanism network (PBATn) to monitor and recognize the daily behavior of pangolins, including their breeding behavior. PBATn consists of a backbone and a PBATn head [29]. The backbone includes the basic ResNet-50 structure [30] with temporal relation, self-attention, and channel attention layers. The temporal relation layer serves to associate temporal information by transferring partial channel weights between the previous and subsequent frame; the self-attention layer extracts key image features by constructing 3 × 3 local and global attention matrices; and the channel attention layer adjusts the importance of weights by setting different weights for channels [31]. The data frame extracted by the backbone features is input into the PBATn head. The PBATn head uses a consensus function to fuse class weights, and outputs behavior scores and classification results through a classifier. The network structure is shown in Figure 2.

#### 2.5.1. Backbone

The backbone, an improvement upon Resnet-50 (Figure 2), consists of five large convolutional modules: conv1, conv2_x, conv3_x, conv4_x, and conv5_x (where x denotes the number of residual blocks in each convolutional module). Eight consecutive 414 × 240 resolution images are compressed into an 8 × 3 × 224 × 224 matrix as input. Conv1 comprises a single 7 × 7 convolution layer and a max pooling layer for subsampling, preserving original image information. The 7 × 7 convolution layer, with a stride of 2, yields 64 channels. Conv2_x, conv3_x, conv4_x, and conv5_x each contain three, four, six, and three residual blocks, respectively, to mitigate gradient vanishing. The first residual block in each module (conv2_1, conv3_1, conv4_1, conv5_1) is a convolution block (orange in Figure 2) used for channel and feature map adjustments, while the subsequent blocks are identity blocks (yellow in Figure 2) deepening feature learning. Both block types share a common structure, differing in that the convolution block includes an additional downsampling layer and channel conversion in the secondary path. The downsampling layer, composed of a 1 × 1 convolution layer, has a stride of 1 for conv2_1 and a stride of 2 for conv3_1, conv4_1, and conv5_1.

To improve the efficiency of extracting pangolin behavior features, the original ResNet-50 residual block’s main path is enhanced as follows: one time-series relation module, one 1 × 1 convolution, one self-attention module, one 1 × 1 convolution, and one channel attention module (CA Layer). These additions are integrated into the main path of the residual block to extract beneficial features for behavior classification and minimize the pangolins’ cage background impact. The time-series relation layer focuses on temporal information across frames and enhances network recognition speed. It comprises transfer layers that pass 1/8 of the current frame’s channel features to the next frame, and inherits 1/8 of the previous frame’s channel features to establish time-series feature relationships. The self-attention module replaces the original 3 × 3 convolution layer in ResNet-50. It utilizes the attention key to redefine feature values, key, query, and value, effectively limiting computation and network parameter consumption while promoting the learning of key image features. Key values are derived from a 3 × 3 convolution, producing attention matrices (K1, K2, K3) to identify feature relationships and calculate weighted averages. The Channel Attention Layer consists of an average pooling layer, two Linear layers, and two activation function layers. It assigns varying weights to image channels to reinforce essential feature information. The average pooling layer utilizes the AvgPool function to obtain global receptive field information. Linear layers output feature weights, followed by ReLU and Sigmoid activation functions. These layers modify the feature map Y by multiplying it with the Sigmoid layer’s output feature map. Additionally, an adaptive average pooling layer (AvgPool) is inserted after the last residual block of conv5_x to preserve high-dimensional information.

The network was trained with a small batch size in the 5 convolutional modules, and each 1 convolutional layer (7 × 7 and 1 × 1 convolutional layers) was followed by a GN layer and a ReLU layer. Group Normalization (GN) is a method for neural networks that normalizes input data by dividing it into multiple groups and calculating the mean and variance of each group. The GN layer consists of 1 GroupNorm function, with the group set to 32 groups, meaning input channels are divided into 32 groups and normalized by group calculation. The advantage of the GN layer is that it can better preserve the statistical properties of the data. Compared to traditional normalization methods, it is more flexible and does not require all the data to be pulled to a fixed distribution, but instead allows each group to have its own distribution; thus, better capturing the structural information of the data. Additionally, the computation complexity of Group Normalization is lower because it only needs to calculate the mean and variance of the groups, not the mean and variance of all data. The ReLU layer contains 1 ReLU activation function. The advantage of the ReLU layer is that it is computationally fast and can prevent neuron “death.” The “death” refers to the phenomenon of a neuron outputting 0, which can cause the entire neural network to be unable to train. The ReLU layer effectively avoids this by setting negative input values to 0 and preserving positive input values unchanged. The output of the ReLU layer has a non-linear feature, which is helpful in extracting non-linear features.

#### 2.5.2. PBATn Head

The PBATn head consists of 1 average pooling layer, 1 dropout layer, 1 FC (Fully Connected) layer, 1 feature fusion function (consensus), and 1 loss function. The average pooling layer and FC layer form the category classifier, receiving 2048 channel weights from the output of the backbone, which results in 6 feature weights corresponding to 6 behavior categories. A dropout layer is added before the FC layer to discard features that are not related to the 6 behaviors. The feature fusion function is responsible for fusing the behavior category weights of 8 frames of images and outputting the behavior category score. The loss function (cross-entropy loss) uses the standard cross-entropy loss function for classification. The cross-entropy loss function can speed up the convergence of the network and update the weight matrix; its loss rate is only related to the probability of the correct category, reducing the computation of the network. The loss function is shown in Equation (1) [32].
(1)$$L(y_{i},y\prime_{i})=-\sum_{i∊|C|}^{i}y_{i}logy\prime_{i},$$
where *y*_i_ represents the true value of category *i*, y′_i_ represents the predicted value of category *i*, and *C* represents the total number of categories.

#### 2.5.3. PBATn Improvement

There are two major improvements to the PBATn network. (1) The temporal relation, self-attention, and channel attention layers are added to the residual block in the main path structure of ResNet-50. A temporal relation layer is inserted before the main path of the residual block to associate the temporal information of the previous and current frames. The 3 × 3 convolutional layer in the middle of the residual block is replaced by a self-attention layer, which generates the self-attention feature value Y through the key, query, and value, reflecting the focus intensity of different spatial locations in a single frame image. After the main path of the residual block, a CA layer, which assigns different weights to image channels and highlights the channel information with the highest relevance to the pangolin feature, is inserted.

(2) A GN layer and a ReLU layer are added after each 7 × 7 and 1 × 1 convolutional layer in ResNet-50. The GN layer consists of 1 GroupNorm function, which divides the input channels into 32 groups and then normalizes each group, preserving the distribution features of each group and reducing the computational complexity. The ReLU layer contains 1 ReLU activation function, which increases the nonlinearity of the network, is computationally fast, and can prevent gradient disappearance.

### 2.6. Network Workflow

Each video was divided into 8 segments during sample training, and 1 frame was randomly selected from each segment; thus, a total of 8 frames were input. The size of a single-frame image is 3 × 414 × 240 (number of channels × frame length × frame width). In actual use, video was obtained online, and images were captured using a sliding window method. The frame-sampling time interval was set according to the frame rate, such as 30 fps/s, and then 1 frame was randomly selected every second. After 8 frames were collected, they were used as input. The size of each input frame image was transformed by scaling the original image with the same aspect ratio to a length of 224, and then the width was padded with [0,0,0] pixels to a square shape. The size after transformation was 3 × 224 × 224 (number of channels × frame length × frame width).

The transformed images were input into the backbone through the data loader interface, and the forward function was called for calculation. The images passed through conv1, the max pooling layer, conv2_x, conv3_x, conv4_x, conv5_x, and the average pooling layer, with output sizes of 64 × 112 × 112, 64 × 56 × 56, 256 × 56 × 56, 512 × 28 × 28, 1024 × 14 × 14, 2048 × 7 × 7, and 2048 × 1 × 1 feature maps, respectively. The output feature maps were input into the PBATn head, and the category classifier output the feature weights of behaviors such as mounting, chasing, feeding, resting, activity, and others. Then, the fusion function fused the weights of 8 frame images to output the score of each behavior.

### 2.7. Behavior Recognition Evaluation Metrics

In order to evaluate the performance of the algorithmic model, accuracy, recall/true-positive rate (TPR), precision, false-positive rate (FPR), specificity and F1Score [33] model evaluation metrics are used here. (F1Score), as shown in Equations (2)–(7), respectively:(2)$$\mathrm{Accuracy}=\frac{\mathrm{Tp}+\mathrm{Tn}}{{\mathrm{Sum}}_{d}},$$
(3)$$\mathrm{Recall}/\mathrm{TPR}=\frac{\mathrm{Tp}}{\mathrm{Tp}+\mathrm{Fn}},$$
(4)$$\mathrm{Precision}=\frac{\mathrm{Tp}}{\mathrm{Tp}+\mathrm{Fp}},$$
(5)$$FPR=\frac{\mathrm{Fp}}{\mathrm{Fp}+\mathrm{Tn}},$$
(6)$$\mathrm{Specificity}=\frac{\mathrm{Tn}}{\mathrm{Fp}+\mathrm{Tn}},$$
(7)$${F1}_{\mathrm{Score}}=2\frac{\mathrm{Precision}\times \mathrm{Recall}}{\mathrm{Precision}+\mathrm{Recall}}$$
where Tp, Tn, Fp, and Fn represent the number of samples of deer behavior in the true, true negative, false positive, and false negative categories, respectively. sumd represents the total number of samples identified. TPR represents the inverse correlation with FPR. TPR represents sensitivity; the higher the TPR, the higher the probability of selecting the correct sample. The higher the FPR, the lower the probability of selecting the correct sample. F1Score represents the value of F1Score and is the value at which both recall and accuracy are at their highest.

Accuracy is used to evaluate the proportion of correctly identified samples out of all the behavioral samples of the pangolin to the total sample. Generally speaking, the higher the correct rate, the better the classification algorithm, but when the samples are unbalanced, relying on the correct rate alone cannot properly evaluate the model. Therefore, it is more objective to evaluate the model using a combination of recall, accuracy, and F1 values. Recall refers to the proportion of true classes among all classes correctly identified by the model, indicating the proportion of behaviors actually correctly identified; accuracy refers to the proportion of true classes among all the classes judged to be positive by the model, indicating how correctly the model identifies a particular class of behavior; F1 value is the summed average of accuracy and recall, with a higher F1 indicating that the test method is more effective.

Two graphical tools, the PR curve and the ROC curve, were used to measure the performance of the pangolin behavior recognition model. The PR curve calculates the recall and accuracy of a sample by reclassifying it after varying the threshold once, and then uses this as a coordinate value to plot on a planar coordinate plot. The PR curve is more sensitive to the sample and measures the ability of a classifier to classify in the face of unbalanced data. The balance point (BEP) corresponds to the calculated F1 value. The receiver operating characteristic curve (ROC) is also known as the sensitivity curve, where each point on the curve reflects the same level of sensitivity. The ROC curve can be used for threshold selection and for comparing different models. The value of the area under the ROC curve (AUC) is commonly used to evaluate the effectiveness of diagnostic models.

## 3. Result

### 3.1. Model Training Process Analysis

By pre-training the PBATn network using transfer learning, the existing base, parameter, and knowledge migration, as well as training time, are reduced. For initialization, the PBATn network loads the ResNet-50 pretrained weights. Recordings of the model training procedure, model accuracy, and loss function are documented using PyTorch’s built-in TensorboardX. For the training set, the PBATn network model has an accuracy of 98.95% and a loss function value of 0.1531; for the validation set, the accuracy is 96.11% and the loss function value is 0.1852, as shown in Figure 3.

### 3.2. Identification Result of PBATn Network

Table 4 shows the results of the PBATn network on the test set, with an average accuracy of 99.17%, mAP of 97.50%, mean recall of 97.55%, mean specificity of 99.53% and mean F1 score of 97.48%. The overall recognition result of the model was good. Among them, the highest evaluation indicators were the three behaviors of feeding, resting and disturbing, which were most clearly characterized in the videos. The chasing behavior usually occurred late in the mounting behavior and lasted for a shorter period of time, resulting in a slightly lower chasing precision rate than the other behaviors.

The ROC curves for the six behaviors are shown in Figure 4. The ROC curves were further calculated using macro-averaging and micro-averaging methods. Macro-averaging first calculates the ROC values for each class behavior and then calculates the arithmetic mean to plot the ROC curve; micro-averaging first calculates the mean of all class true rates and false-positive rates and then constructs the overall ROC curve. From Figure 4, it can be seen that the two breeding behaviors of mounting and chasing converge to the ideal index, and the AUC of each class converges to 1.

The effect of PBATn network recognition is verified from a feature visualization perspective, as shown in Figure 5. From the heat map, it can be observed that the highlighted parts are concentrated in the region where the breeding behavior of the pangolin occurs, confirming that the attention mechanism can effectively capture the breeding behavior of the pangolin.

### 3.3. Comparison Results with Other Networks

To assess PBATn’s performance in pangolin behavior recognition, state-of-the-art 3D convolution-based networks (SlowFast [34], X3D [35], R2 + 1D [36], C3D [37], I3D [38]) and temporal module-based networks (TANet [39], TSN [40] improved, PBAn) were chosen for comparison with PBATn for validation. SlowFast employs fast and slow channels for behavioral recognition. X3D extends time, space, width, and depth dimensions to maintain a large perceptual field. R2 + 1D decomposes 3-dimensional spatiotemporal convolution into separate temporal and spatial convolutions. C3D is constructed using 3D convolution and 3D pooling. I3D utilizes inflated convolution to expand 2D convolution. TANet relies on the temporal adaptive module (TAM) with a temporal convolution and an attention mechanism. TSN operates on video with segmental convolution, replacing the 3 × 3 convolution layer with a self-attentive layer for improved network performance. PBAn is also a network improved by us. PBATn eliminates the temporal relative layer and replaces the segment consensus function with a GRU network in the classification header to produce PBAn. PBAn computes temporal information through the GRU network. 

Each model processes each segment of the video entering the input layer. SlowFast takes one frame every 2 frames as input. SlowFast is divided into slowway and fastway. In slowway, every 5 frames are inputted, while in fastway, every 1 frame is inputted. The pretrained model uses the ResNet3d-50 model. X3D takes one frame every 6 frames as input. The pretrained model uses the X3D model. R2 + 1D takes one frame every 8 frames as input. The pretrained model uses the ResNet-34 model. C3D takes 1 frame every 1 frame as input. The pretrained model uses the C3D model. I3D takes 1 frame every 2 frames as input. The pretrained model uses the ResNet-50model. TANet is divided into 8 segments, and one frame is randomly selected from each segment. The pretrained model uses the ResNet-50 model. TSN is divided into 3 segments, and 1 frame is randomly selected from each segment. The pretrained model uses the ResNet-50 model. PBAn and PBATn follow the same training method. Each frame is resized to 3 × 224 × 224 before being inputted into the backbone.

The trained networks were tested with a test set of 1200 videos. Test results for each network type are represented by a confusion matrix in Figure 6. Analysis of the results in Figure 6 reveals that the recognition accuracy of chasing behavior in each network is lower than other behaviors and is easily misidentified as mounting behavior. It is more difficult to recognize when the pangolins are severely overlapped or obscured. The chasing behavior is extremely short, with some videos only lasting one to two frames and occurring before or after the chasing behavior, making it difficult for the model to identify the chasing behavior. When the video is underexposed or overexposed, the model misidentifies it. The PBATn network correctly identified a total of 1170 videos and 30 videos were not correctly identified, which is the highest percentage of chasing behavior compared to other networks.

The precision for each network was calculated and the results are shown in Figure 7. Each network has a high recognition accuracy of 86.00% or more for activity, resting and other behaviors. For the feeding behavior, the PBATn network had the highest recognition accuracy of 99.50%, while the recognition accuracy of SlowFast and X3D was poorer at 12.50% and 62.00%, respectively. For the two breeding behaviors, chasing and mounting, the PBATn network also achieved the highest recognition accuracy of 94.00% and 98.50%, respectively. Among the results of all the networks, the overall recognition accuracy of chasing behavior was lower than that of other behaviors, followed by feeding behavior and mounting behavior, showing that chasing behavior was the most difficult to recognize. 

The PR curves were used to measure the ability of the 9 networks when classifying unbalanced data, as shown in Figure 8. Points A, B, C, D, E, F, G, H and I were the equilibrium points of the PR curves of the nine networks, corresponding to the F1 value of each network. In the figure, the equilibrium point I of the PBATn network was higher than the other points, proving that the PBATn network was more effective in recognition than the other networks.

The last 1 ReLU layer of the network was mapped in a 2D space as a feature-embedding map to examine how well each network discriminated between the six behaviors, and the results are shown in Figure 9. Each point in the figure represents a video of the test set, and each behavioral category is represented by one color. The higher aggregation of each behavior on its own proves that the PBATn network distinguishes the categories of activity, chasing, mounting, feeding, resting and disturbance behaviors better than other networks.

The paragraph discusses a comparison of various networks in terms of their application performance, particularly focusing on accuracy, speed, parameters, and computation. The results are summarized in Table 5. The mAP of 3D convolutional networks and temporal module networks is similar. However, 3D convolutional networks have more parameters and computation, leading to slower detection speeds. This is because they calculate timing information for each video frame differently, often extracting unnecessary data. The X3D network, while expanding perceptual fields, lacks the temporal dimension, resulting in a lower mAP of 68.83%. TANet in the temporal module-based network achieves higher recognition accuracy but slower processing speed due to inefficient weight information handling. TSN, with an attention mechanism, is 2.50 times faster than TANet in terms of detection speed. PBAn uses a GRU network, increasing parameters and computation. However, it achieves a detection speed 2.16 times faster than TSN and 5.61 times faster than TANet by adding self-attention and channel attention layers. PBATn extracts timing information through channel movement, significantly improving recognition speed without increasing parameters. It is 5.60 times faster than PBAn. PBATn, using the FC layer and segment consensus function, reduces parameters by 8.81 MB compared to PBAn. Its attention mechanism maintains detailed feature extraction, resulting in the highest mAP among all the models at 97.50%. Meanwhile, by comparing model parameters, it is found that having too many parameters can also reduce the generalization ability due to overfitting.

## 4. Discussion

### 4.1. Discussion of Data Processing

The pangolin video dataset contained 11,476 segments, 362,622 total frames and six behaviors, mounting, chasing, feeding, activity, resting, and other, of which mounting and chasing are both breeding behaviors. During the data set annotation process, using the BP method to label helps the network model improve sensitivity to breeding behavior. However, the use of the BP method emphasizes prioritizing breeding behaviors. When breeding behaviors occur, other behaviors may be overlooked.

### 4.2. Discussion of PBATn

In the PBATn, the temporal relation, self-attention, and channel attention modules are added to the residual block in the main path structure of ResNet-50. The precision of all behaviors in Table 4 is above 93%, meeting the practical monitoring requirements for pangolin breeding behaviors. The chasing behavior usually occurred late in the mounting behavior and lasted for a shorter period of time, resulting in a slightly lower chasing precision rate than the other behaviors. In Figure 4, breeding behaviors converge to the ideal index, and the AUC of each class converges to 1. From the heat map in Figure 5, the PBATn can accurately capture pangolins with their behavior. Meanwhile, in Figure 10, the PBATn can capture and identify pangolin behavior in different scenarios. This can prove that the attention mechanism can effectively capture the breeding behavior of the pangolin.

### 4.3. Discussion of Different Networks

To assess PBATn’s performance in pangolin behavior recognition, state-of-the-art 3D convolution-based networks (SlowFast [34], X3D [35], R2+1D [36], C3D [37], I3D [38]) and temporal module-based networks (TANet [39], TSN [40] improved, PBAn) were chosen for comparison with PBATn for validation.

In Figure 6, Figure 7, Figure 8 and Figure 9, the results demonstrate the PBATn’s excellent recognition performance and clustering degree. But the recognition accuracy of chasing behavior in each network is lower than other behaviors and is easily misidentified as mounting behavior. It is more difficult to recognize when the pangolins are severely overlapped or obscured. The chasing behavior is extremely short, with some videos only lasting one to two frames and occurring before or after the chasing behavior, making it difficult for the model to identify the chasing behavior. When the video is underexposed or overexposed, the model misidentifies it.

In Table 5, compared to other networks, PBATn demonstrates a higher mAP (97.50%), faster recognition speed (11.79 ms/frame), and fewer parameters (22.70 MB) and FLOPs (40.62 G). In practical application, the increased speed and accuracy of PBATn have enhanced the effectiveness of pangolin behavior recognition, thus validating the reliability of the attention mechanism. However, it is important to note that behavior recognition experiments for pangolins have thus far been conducted solely in artificial captive environments. There is a lack of outdoor pangolin behavior data, and the network’s reliability will be further substantiated by capturing a substantial amount of outdoor behavior data in future studies.

## 5. Conclusions

In this paper, a pangolin breeding attention and temporal relative network (PBATn) was employed to monitor and identify the breeding and daily behaviors of pangolins. Two key improvements were implemented. (1) The temporal relation, self-attention, and channel attention layers effectively addressed the occlusion problem of breeding behavior, resulting in enhanced recognition accuracies for both breeding and daily behaviors. (2) GN and ReLU layers were introduced after each 7 × 7 convolutional layer and 1 × 1 convolutional layer in the ResNet-50 network, aiming to preserve data distribution characteristics, reduce computational complexity, accelerate computation, and prevent gradient disappearance. The dataset comprised 11,476 videos, divided into training, validation, and test sets. The model is suitable for a 2.40 m × 2.20 m (length × width) pangolin cage area, with a nest box measuring 40 cm × 30 cm × 30 cm (length × width × height) positioned either on the left or right side inside the cage. A spherical night-vision monitoring camera was installed on the cage wall at a height of 2.50 m above the ground. On the test set, PBATn surpassed SlowFast, X3D, and others, with superior performance metrics: mAP (97.50%), average accuracy (99.17%), recall (97.55%), specificity (99.53%), and F1 score (97.48%). In the aforementioned scenario, a faster and higher recognition speed of PBATn improved the accuracy of pangolin behavior recognition, proving the reliability of the attention mechanism. This study shows that the deep learning system can accurately observe pangolin breeding behavior and it will be useful for analyzing the behavior of these animals.

## Figures and Tables

**Figure 1 animals-14-01032-f001:**
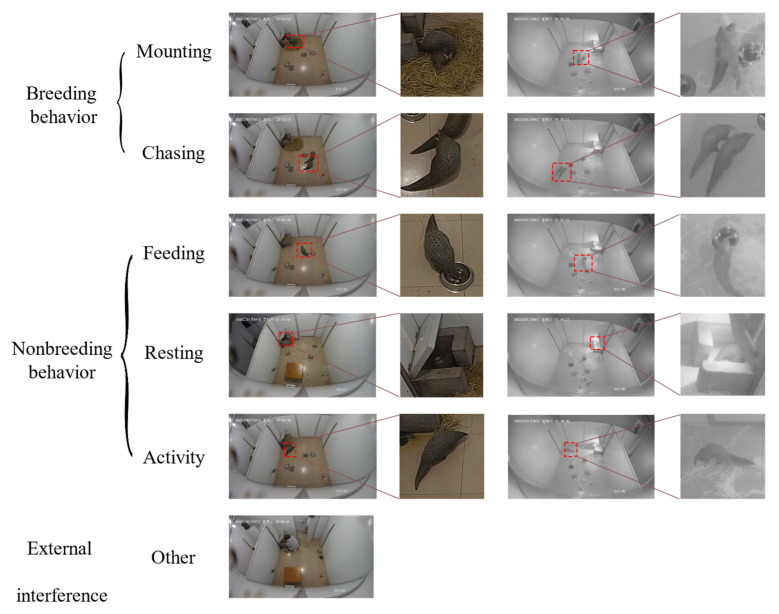
The labels of the pangolins’ behavior. The left and right columns are the corresponding image frames of the same behavior intercepted from daytime and nighttime videos, and the right small image is the display after enlarging a specific area (red frame) where a behavior is observed in an image frame.

**Figure 2 animals-14-01032-f002:**
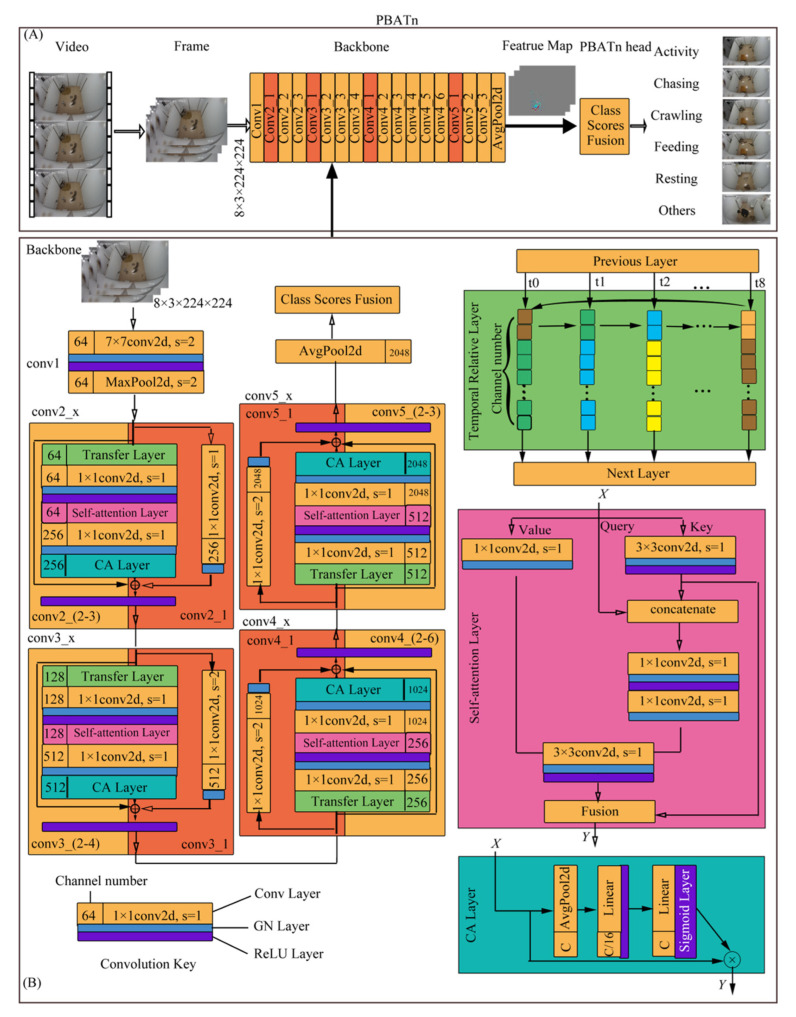
The diagram draws the structure of PBATn. (**A**) represents the overall structure and processing data flow of PBATn, while (**B**) represents the internal structure and processing data flow of the backbone.

**Figure 3 animals-14-01032-f003:**
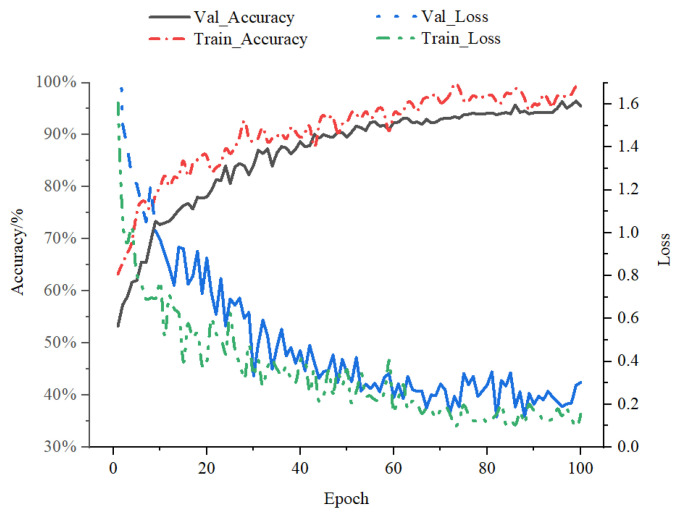
Recognition accuracy and loss value of PBATn. The loss function value lowers and the accurate rate of the training set gradually rises. The validation set’s correct rate and loss function value agree well with the equivalent value from the training set.

**Figure 4 animals-14-01032-f004:**
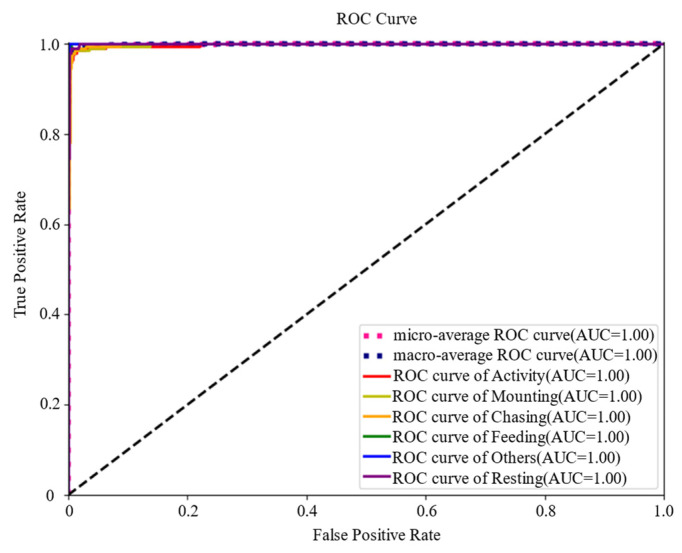
ROC curve. The ROC curves are all located at the top left of the random guess line; the closer the curve is to the top left corner, the more accurate the classification result is shown.

**Figure 5 animals-14-01032-f005:**
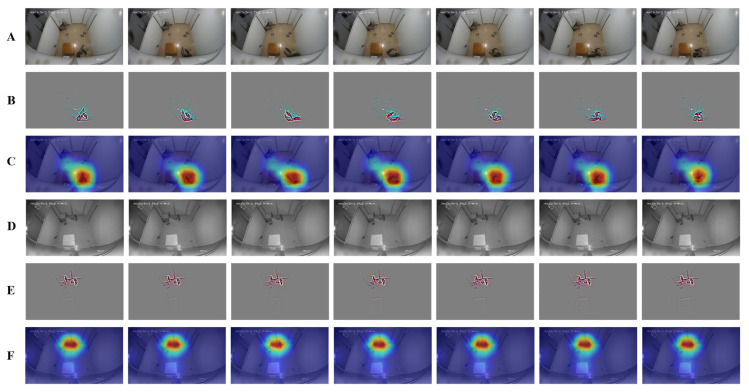
Feature visualization: Rows (**A**,**D**) depict consecutive frames of breeding behavior from the test set, with (**A**) as a color frame and (**D**) as a black-and-white frame. Rows (**B**,**E**) display the results of the feature visualization output from the last convolution layer of the network model. Rows (**C**,**F**) represent heat maps generated by the feature attention mechanism based on (**B**,**E**), respectively, with warmer colors indicating more intense breeding actions.

**Figure 6 animals-14-01032-f006:**
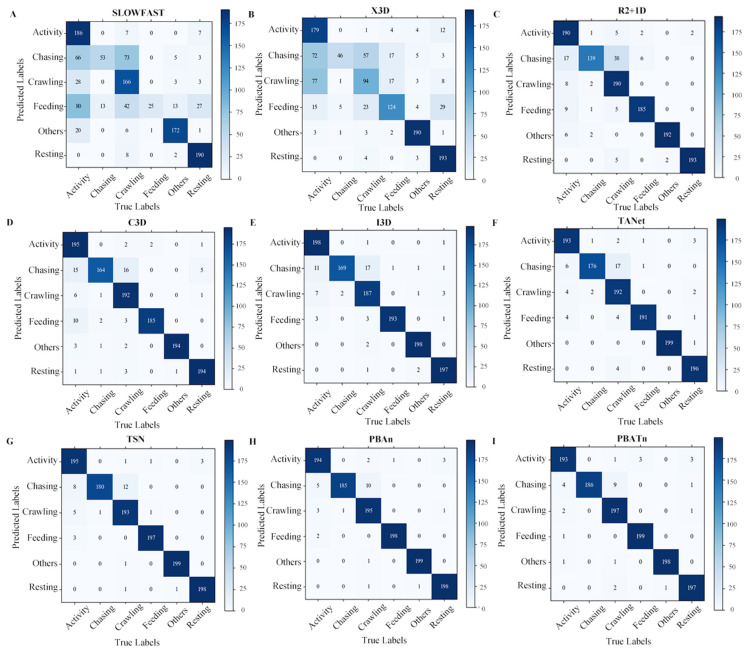
(**A**–**I**) respectively represent the Confusion matrixs of SlowFast, X3D, R2 + 1D, C3D, I3D, TANet, TSN, PBAn, and PBATn. The trained networks were tested with a test set of 1200 videos, with 200 videos for each behavior.

**Figure 7 animals-14-01032-f007:**
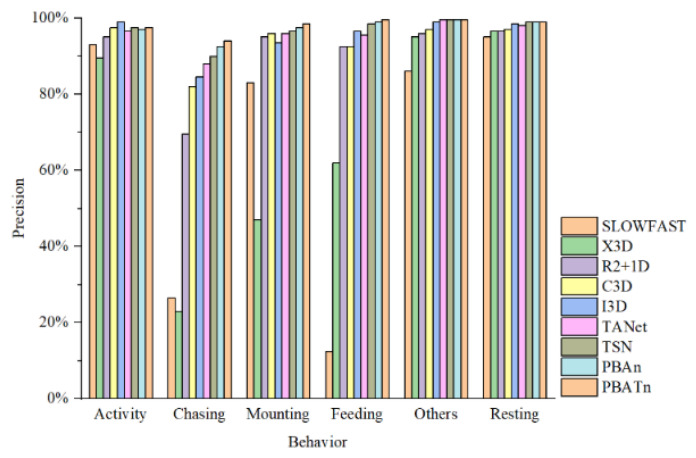
The precision rate for each network and each behavior was calculated.

**Figure 8 animals-14-01032-f008:**
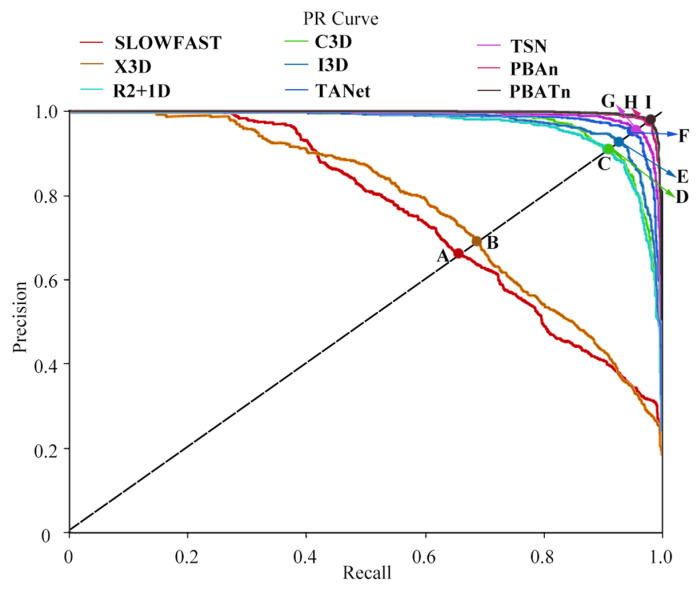
Points A, B, C, D, E, F, G, H and I were the equilibrium points of the PR curves of the nine networks, corresponding to the F1 value of each network. The closer the PR curve is to the top-right corner, the more accurate the classification result is shown.

**Figure 9 animals-14-01032-f009:**
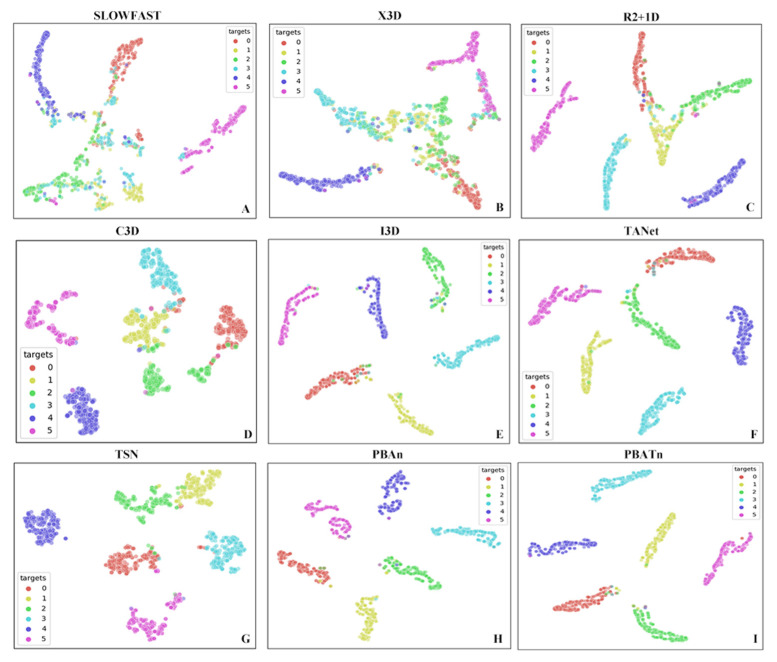
(**A**–**I**) respectively represent the feature embeddings of SlowFast, X3D, R2 + 1D, C3D, I3D, TANet, TSN, PBAn, and PBATn. The category 0 (activity), 1 (chasing), 2 (mounting), 3 (feeding), 4 (rest), and 5 (disturbance) feature mapping points are represented by red, yellow, green, blue, purple, and pink, respectively; the different behaviors of the PBATn network are clustered in six regions, and each behavior has its own clustering degree.

**Figure 10 animals-14-01032-f010:**
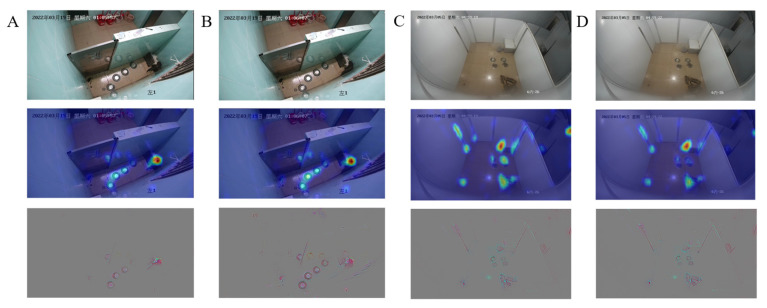
Displays the PBATn’s recognition results using a heatmap. (**A**,**B**) represent pangolins exhibiting mounting behavior in the old enclosure. (**C**,**D**) demonstrate recognition results of mounting behavior in scenarios different from Figure 4.

**Table 1 animals-14-01032-t001:** Data collection with pangolin id and batch.

Batch	18–19 Aug. 2021	16–18 Sept. 2021	07–10 Jan. 2022	02–07 Mar. 2022	17–21 Mar. 2022
Pangolin id	Z1	M5	Z2, Z6	Z2, Z6, Z9	Z8, Z12, M13, M21

**Table 2 animals-14-01032-t002:** Video behavior division.

Behavior Label	Number of Video Segments	Number of Video Frame
activity	3367	106,981
chasing	544	17,190
mounting	2999	95,251
feeding	642	19,930
resting	3274	104,759
other	660	19,511
total	11,476	363,622

The proportion of mounting, chasing, feeding, activity, resting, and other in the dataset is 7:1:1:7:7:1.

**Table 3 animals-14-01032-t003:** Data set of pangolins.

DataSet	Number of Video Segments	Number of Video Frame	Total Time (min)
training set	8221	260,480	4341.32
validation set	2055	65,120	1085.33
test set	1200	38,022	633.70
total	11,476	363,622	6060.35

The number of image sets of each behavior in the test set was 200, for a total of 1200 image sets, accounting for 10% of the overall image sets. The remaining 90% of the image sets were randomly divided into a training set and validation set at a 4:1 ratio.

**Table 4 animals-14-01032-t004:** Performance statistics of PBATn (%). Different metrics for the six behaviors of pangolins were evaluated.

PBATn	Accuracy	Precision	Recall	Specificity	F1_Score
activity	98.75	96.50	96.00	99.30	96.20
chasing	98.83	93.00	100.00	98.80	96.37
mounting	98.67	98.50	93.80	99.70	96.10
feeding	99.67	99.50	98.50	99.90	99.00
resting	99.75	99.00	99.50	99.80	99.20
others	99.33	98.50	97.50	99.70	98.00
Average	99.17	97.50	97.55	99.53	97.48

**Table 5 animals-14-01032-t005:** Comparison of networks in results. The mAP, processing speed, params and flops of each network are evaluated comprehensively.

Models	mAP/%	ms/Frame	Params/MB	Flops/G
SlowFast [34]	66.00	526.31	33.57	49.42
X3D [35]	68.83	89.28	2.99	17.62
R2 + 1D [36]	90.75	800.00	63.56	286.19
C3D [37]	93.67	666.67	54.21	63.11
I3D [38]	95.17	202.02	27.24	58.58
TANet [39]	95.58	370.37	24.78	51.04
TSN (improved) [40]	96.83	142.86	23.52	51.03
OUR (PBAn)	97.00	66.01	31.51	40.62
OUR (PBATn)	97.50	11.79	22.70	40.62

## Data Availability

All data presented in the study are available under reasonable request from the corresponding author.

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
