# Peer review of "Automatic Identification of Pangolin Behavior Using Deep Learning Based on Temporal Relative Attention Mechanism"

_animals, 2024, doi:10.3390/ani14071032_

Round 1

Reviewer 1 Report (Previous Reviewer 1)

Comments and Suggestions for Authors

Acoording to the previous version, this manuscript is perfect.

Author Response

Thanks

Reviewer 2 Report (Previous Reviewer 3)

Comments and Suggestions for Authors

The authors have made the revisions I requested. The conclusions is now limited to pangolins. The original paper had overstated the ability of the computer system. The paper is now acceptable for publication.

Comments on the Quality of English Language

N/A

Author Response

Thanks

Reviewer 3 Report (Previous Reviewer 2)

Comments and Suggestions for Authors

The author did not address the review comments from the last submission:

"The author presented in the paper a method to automatically recognize the breeding and daily behaviors of pangolins based on video data using a deep neural network. It is a good topic to use deep neural networks in solving complex real-world problems, however, the experiment setup in this paper is far from real-world scenarios. The data in the whole experiment came from a tiny scene (2.40 m × 2.20 m) recorded by a fixed camera, therefore the variety of the data is extremely poor. This leads to a high similarity between the test set and training set, therefore it is very likely that the reported result is an overfit. The work is not ready for publication because the experiment cannot prove the proposed method has any significance for real applications."

As the experiment is the same as the previous version, I will not change the recommendation. Here are some further comments on the experiment:

1. The paper did not mention the position of the resting box, and from the images provided, I assume it was never moved. Therefore resting recognition may be highly correlated to the position rather than behavior. The author should include a test with a different room layout.

2. The paper did not mention anything about augmentation. Given the simplicity of the training data and the large scale of the network, it is highly likely that the network overfitted to the scene. The author should at least include an experiment with mirrored video or other augmentations to demonstrate the network has some generalization power.

Comments on the Quality of English Language

The English is ok.

Round 2

Reviewer 3 Report (Previous Reviewer 2)

Comments and Suggestions for Authors

The authors tested the model in two new environments in the discussion, which showed some generalization power. However, this does not strengthen the conclusion of the paper (line 27-32, line 513-516). As we mentioned earlier, all the data collected for the experiments are highly similar; this both affects training (low entropy; larger models will overfit) and evaluation (cannot show generalization power). To make the paper publishable without redesigning the experiment, the authors should use precise scopes when describing the results. This includes but is not limited to:

1. In the abstract, describe the camera, lighting, and environment setup before showing the result and comparing it with other models. Currently, the abstract does not mention when and where the method is applicable.

2.  In section 3.3 (Comparison results with other networks), add details on how other networks were trained, including how the data was adapted for the input layer and whether pre-trained weights were used. Compare the model sizes and mention the fact that having more parameters (line 442) could also decrease the generalization power because of overfitting, which is a possible explanation for performance differences.

3. In the conclusion, similar to the abstract, mention the scope of which the method is applicable, including camera, lighting, environment setup, etc., and the method is only proven to be better than other established methods under these constraints.

Comments on the Quality of English Language

The writing needs improvements. Some sentences are confusing. Please proofread the manuscript carefully, especially the modified sections.

Author Response

This manuscript is a resubmission of an earlier submission. The following is a list of the peer review reports and author responses from that submission.

Round 1

Reviewer 1 Report

Comments and Suggestions for Authors

1. Line 38: “0. Introduction” The numbers should start with 1, please correct all section captions.

2. Line 263: There is no y(hat) in the model. Please correct the model writing.

3. Lines 315-347: The sub-section “2.2 Behavior recognition evaluation index” is not a part of results. So it should be moved to the Methods section.

4. Line 318: “Specificity” was given two times. Please delete one of them.

5. Line 373: Figure 5 is too small to recognize the images. You can give less figures with larger sizes.

6. Line 405: Figure 6 is too small and its quality is low. Not readable even enlarged.

The Introduction section was written well detailed and the flow of expressions was very orderly. Animal material and methods used was introduced well.

The Materials and Methods section was described detailed and easy to understand.

The Results and Discussion section should be improved especially for image quality. There is no discussion. It should be written.

The Conclusion section is not enough. It is written at a very simple level for this work that has been done with a lot of effort. It does not reflect the content.

Reviewer 2 Report

Comments and Suggestions for Authors

The author presented in the paper a method to automatically recognize the breeding and daily behaviors of pangolins based on video data using a deep neural network. It is a good topic to use deep neural networks in solving complex real-world problems, however, the experiment setup in this paper is far from real-world scenarios. The data in the whole experiment came from a tiny scene (2.40 m × 2.20 m) recorded by a fixed camera, therefore the variety of the data is extremely poor. This leads to a high similarity between the test set and training set, therefore it is very likely that the reported result is an overfit. The work is not ready for publication because the experiment cannot prove the proposed method has any significance for real applications.

Comments on the Quality of English Language

English is ok. There are some grammatical errors that do not impair the meaning of the article.

Reviewer 3 Report

Comments and Suggestions for Authors

The methods used for training the deep learning program need to be more completely described. The conclusions overstate the ability to improve pangolin breeding success.

Line 17 - Change the last sentence to "This study shows that the deep learning system can accurately observed pangolin breeding behavior and it will be useful for analyzing the behavior of these animals."

Lines 33-35 - Change the last sentence of the abstract to the same as line 17.

Line 145 - You need to completely explain how the training set was made. This reviewer assumes that human observers had to input the initial data to train the deep learning system. Please describe exactly how you did this.

Line 158 - The pictures need to be made bigger. When viewed with a magnifying glass, they were blurred and the animals were difficult to see.

Line 166 - Were the intercepted video segments viewed by people?

Lines 473-475 - The conclusions overstate your results. Change to "Our study showed that the deep learning system can easily quantify breeding behavior. This will be really useful for wildlife scientists working with pangolins."

Comments on the Quality of English Language

It is fine.